# High-Throughput Computational Screening of Two-Dimensional Covalent Organic Frameworks (2D COFs) for Capturing Radon in Moist Air

**DOI:** 10.3390/nano13091532

**Published:** 2023-05-03

**Authors:** Hongyan Zeng, Xiaomin Geng, Shitong Zhang, Bo Zhou, Shengtang Liu, Zaixing Yang

**Affiliations:** 1State Key Laboratory of Radiation Medicine and Protection, School for Radiological and Inter-Disciplinary Sciences (RAD-X), Collaborative Innovation Center of Radiation Medicine of Jiangsu Higher Education Institutions, Soochow University, Suzhou 215123, China; hyzeng@stu.suda.edu.cn (H.Z.); xmgengxmgeng@stu.suda.edu.cn (X.G.); 2State Key Laboratory of Separation Membranes and Membrane Processes, School of Chemical Engineering and Technology, Tiangong University, Tianjin 300387, China; zhangshitong@tiangong.edu.cn; 3School of Big Data and Artificial Intelligence, Chengdu Technological University, Chengdu 611730, China; bozhou@zju.edu.cn

**Keywords:** radon capture, two-dimensional covalent organic frameworks, virtual screening, structure–performance relationship, GCMC simulations

## Abstract

Radon (Rn) and its decay products are the primary sources of natural ionizing radiation exposure for the public, posing significant health risks, including being a leading cause of lung cancer. Porous material-based adsorbents offer a feasible and efficient solution for controlling Rn concentrations in various scenes to achieve safe levels. However, due to competitive adsorption between Rn and water, finding candidates with a higher affinity and capacity for capturing Rn in humid air remains a significant challenge. Here, we conducted high-throughput computational screening of 8641 two-dimensional covalent organic frameworks (2D COFs) in moist air using grand canonical Monte Carlo simulations. We identified the top five candidates and revealed the structure–performance relationship. Our findings suggest that a well-defined cavity with an approximate spherical inner space, with a diameter matching that of Rn, is the structural basis for a proper Rn capturing site. This is because the excellent steric match between the cavity and Rn maximizes their van der Waals dispersion interactions. Additionally, the significant polarization electrostatic potential surface of the cavity can regulate the adsorption energy of water and ultimately impact Rn selectivity. Our study offers a potential route for Rn management using 2D COFs in moist air and provides a scientific basis for further experimentation.

## 1. Introduction

Radon (Rn) is an incredibly harmful air pollutant to the human body, even though its concentration in the atmosphere is a mere 6 × 10^−11^ ppb [1]. This is due to the continuous emission of radioactive Rn from soil, rocks, and building materials that contain radioactive uranium or thorium, leading to increased concentrations of Rn in enclosed spaces such as basements, mines, and housing buildings [2]. The resulting Rn and its decay products are recognized as the primary contributors of natural ionizing radiation, posing a significant risk of lung cancer [3]. As a result, there is a global focus on studying the capture of radon from the atmosphere [4,5,6]. To date, there are three primary strategies for removing radon from confined spaces: ventilation systems [7], plugging diffusion paths [8], and adsorption methods [9,10]. Among them, the use of porous materials as solid adsorbents in adsorption methods is the most cost-effective and technically feasible approach.

Coconut-activated charcoal (AC) is currently the most effective commercialized solid Rn adsorbent, but it also has significant drawbacks. Its ability to maintain a stable and high Rn adsorption level under conditions of high humidity is limited, restricting its widespread use in Rn-rich scenarios with high air humidity, such as water treatment plants, basements, and underground mines. Moreover, due to the lower distribution of Rn size-matching cavities, AC only exhibits moderate Rn capture efficiency and cannot remove Rn thoroughly [11,12]. Interestingly, the metal–organic framework (MOF) has emerged as an alternative resource for this application. Recent virtual screening conducted by Zeng et al. [13] on 23 different kinds of representative MOFs using grand canonical Monte Carlo (GCMC) simulations under completely dry conditions revealed that the zeolitic imidazolate framework (ZIF)-12 is a promising candidate for capturing Rn, with the most outstanding Rn selectivity and adsorption capacity over other candidates. However, it is unclear whether or not humidity affects its performance. Recently, a computation-ready, experimental (CoRE) ZIF structural database covering 48 different structures with 19 topologies was built, and GCMC simulations were conducted on the database [14]. The virtual screening results demonstrate that ZIF-7 exhibits the most stable and balanced performance in Rn selectivity and adsorption capacity under different air humidities. Furthermore, guided by molecular dynamics simulations, extensive modifications have been made to ZIF-7 to make its Rn capture function kinetically feasible. However, owing to the existence of metal nodes in MOFs, the majority of MOFs exhibit a relatively high synthesis cost and sensitivity to humidity in terms of their structural stability [15,16]. Therefore, there is an urgent need to identify more economical and hydrolytic stability candidates for Rn capture.

In recent years, covalent organic frameworks (COFs) have seen immense popularity as typical porous materials, owing to their unique properties and rapid development [17,18,19]. They are formed by organic linkers through covalent bonds, which can be classified into two-dimensional (2D) and three-dimensional (3D) COFs, depending on the type of conjugated building blocks utilized. The exceptional structure of COFs provides them with distinct advantages such as low density, permanent porosity, adjustable pore size, large specific surface area, good chemical and thermal stability [20,21]. Consequently, COFs have found wide applications in the separation/capture of gas pollutants [22,23]. Notably, 2D COFs possess ordered periodic skeletons and inherent polygonal pore connections that form dense one-dimensional pore channels. Furthermore, the one-dimensional open channels with uniformly small windows and no intersections or narrow junctions can prevent unfavorable path blockage and pore blockage [24]. Thus, one-dimensional COFs with these characteristics allow the easier realization of the construction of porous materials with well-defined pores, discrete pore diameters, and polygonal shapes. Recent studies by Yuan et al. [25] and Tong et al. [26]. have demonstrated that 2D COFs exhibit exceptional separation performance for the noble gases Xe and Kr. However, the efficiency of COFs for capturing the noble gas Rn from moist air has not been fully explored, and the underlying structure–performance relationship remains poorly understood.

In this study, we investigated the performance of 8641 2D-layered COFs in the selective adsorption of radon through GCMC simulations. Our results allow us to identified five top-performing candidates with stable and balanced performance in terms of Rn selectivity and adsorption capacity in moist air. Additionally, we reveal the structure–performance relationship of 2D COFs and, importantly, uncover the molecular mechanism by which humidity affects Rn capture performance. These findings not only present a new avenue for Rn capture using 2D COFs but also provide a scientific basis for designing efficient radon trapping materials under high-humidity conditions.

## 2. Methods

### 2.1. COF Database

Mercado et al. [27] compiled a database of 69,840 COFs, mostly novel and assembled in silico, from 666 distinct organic linkers and four established synthetic routes. This comprehensive database includes 8641 2D COFs and 61,199 3D COFs. For the purposes of our study, we focused on the 2D COF database, which consists of 8641 COFs [27]. A 2D COF is a framework comprising stacked 2D layers held together by dispersion interactions [27]. We adopted the nomenclature for COFs in accordance with Berend et al.: linkerA_linkerB_net, where linkerA and linkerB denote the linkers and their respective linker terminals, and net denotes the topology [27]. The corresponding International Union of Pure and Applied Chemistry (IUPAC) names for all linkages and their three-dimensional representations are listed in a previous publication [28] and partially in Appendix A.

### 2.2. Computational Details

To evaluate the Rn capture potential of COFs, we conducted GCMC simulations using a moist air mixture of Rn/N_2_/O_2_/H_2_O with molar ratios of 0.001/0.776/0.209/0.014, at 298 K and 1 bar. Additionally, we investigated Rn adsorption under various total pressures (10–140 kPa) and humidities (relative humidity, RH, of 0.0–1.0), with different Rn concentrations (molar ratio of Rn, *X_Rn_* = 0.00001, 0.00005, 0.0001, 0.0005, 0.001). It is worth noting that the concentration of Rn in the air can differ significantly depending on the location, with values ranging from ~10^−11^ ppb in common places to ~10^−8^ ppb in caves or poorly ventilated basements and houses [29,30,31,32]. Nevertheless, the concentration of Rn is significantly lower than that of other major air components such as N_2_, O_2_, and H_2_O. Consequently, higher Rn concentrations were conventionally utilized in the GCMC simulations to ensure the comparability of Rn uptake performance among different materials.

The GCMC simulations were performed using the RASPA software (version 2.0), as described in a previous study [33]. The Lennard-Jones (LJ) parameters of COFs were obtained from the universal force field (UFF) [34]. The atomic charges were assigned using the charge equilibration method (Qeq) [35], which is widely used in high-throughput screening for the performance of gas separation of porous materials [36]. The N_2_ and O_2_ adsorbates were modeled with the three-site molecular model [37,38,39]. The Tip3P water model was used for water molecules [40]. The Rn model used here was developed by Sanjon et al. [41], based on two primary reasons. First, this model can accurately reproduce both the gas density of Rn and its physical properties when interacting with water molecules, which was essential for our calculations under humid conditions. Additionally, the this model has already been proven to have reliable predictability in terms of the Rn capturing performance of MOFs [14]. The force field parameters for the adsorbate molecules (e.g., Rn, N_2_, H_2_O, and O_2_) are listed in Table 1, which includes the geometric and LJ potential parameters, as well as the atomic charges. The cross LJ interactions were treated with the Lorentz−Berthelot combining rules, and a cut-off distance of 12 Å was used to truncate the LJ interactions. The long-range electrostatic interactions were calculated using the Ewald summation method [42]. Configurations were visualized using the VMD software [43,44]. The surface electrostatic potential of the COFs was calculated using the adaptive Poisson–Boltzmann solver [45,46]. In addition, the diameters and volumes of Rn capturing cavities in COFs were computed using the Multiwfn program [47]. Initialization was achieved after 5000 cycles, while the overall average gas adsorption level was obtained after 5000 cycles.

The adsorption of radon obtained from GCMC simulation is the absolute adsorption amount ($N_{Rn}$). In the separation process, adsorption selectivity measures an important parameter of the separation capacity of the material. In gas mixtures, the selectivity of gas *i* with respect to gas *j*, *k* and *m* is defined as
(1)$$S_{\mathrm{abs}}\left(i/(j+k+m)\right)=\frac{x_{i}/{(x}_{j}+x_{k}+x_{m})}{y_{i}/(y_{j}+y_{k}+y_{m})}$$
where *x* and *y* are the molar ratios of adsorbed species in the adsorbed and bulk phases, respectively [48]. In addition, a metric named adsorbent performance score (APS) [49] was introduced to evaluate the integrated capture performance of radon, which is defined as
(2)$${APS}_{Rn}=N_{Rn}\times S_{abs}\left(Rn/(N_{2}+H_{2}O+O_{2})\right)$$

## 3. Results and Discussion

### 3.1. High-Throughput Computational Screening of Rn-Capturing 2D COFs

Moisture is a critical factor that affects the adsorption of noble gas by porous materials [14]. Thus, in this study, we considered the Rn capture capability of 2D COFs from a quaternary mixed gas (Rn/N_2_/O_2_/H_2_O) to mimic radon exposure in moist air. To assess the performance of the adsorbent materials, we adopted two indexes, Rn selectivity and Rn uptake capacity, as they are crucial for conducting cost-effective gas separation processes [20].

Appendix A depicts the relationship between Rn selectivity and capacity and the increasing selectivity and capacity of 2D COFs for N_2_ and O_2_, respectively. It is evident from Appendix A that COFs with a Rn selectivity of above 100 have very low selectivity for nitrogen and oxygen molecules, ranging from 0.0–0.3 and 0.05–0.70, respectively. Similarly, COFs with high Rn uptake capacity exhibit relatively low nitrogen and oxygen uptake capacity. Interestingly, the Rn selectivity shows a negative correlation with the water selectivity, with COFs that have high Rn selectivity exhibiting extremely low selectivity to water (Figure 1a). Similarly, COFs with high Rn uptake capacity demonstrate a relatively low water uptake capacity (Figure 1b). Therefore, among the three gas components, water has the most significant influence on the selective capture of radon. Figure 1c shows the adsorption performance of all COFs for Rn, with blue, green, and red colors representing COFs with low (<50 mol/kg), medium (50 to 190 mol/kg), and high (>190 mol/kg) *APS_Rn_*, respectively. Based on the adsorption performance results, we screened the top five COFs with the highest *APS_Rn_* values (>190 mol/kg), including linker105_C_linker13_C_kgm (~300.85 mol/kg) > linker101_C_linker11_C_kgm (~287.26 mol/kg) > linker107_C_linker13_C_kgm (~215.24 mol/kg) linker99_C_linker11_C_kgm (~198.26 mol/kg) > linker99_C_linker13_C_kgm (~193.60 mol/kg). The IUPAC names and structures of the linker groups in the top five COFs can be found in Appendix A. Additionally, we tested another set of Rn force field parameters developed by Mick et al. (Appendix A) [50] to assess the Rn capturing performance order of the five candidates. The results showed that the obtained Rn capturing performance order was in good agreement between the two force fields (Appendix A). Moreover, the measured Rn selectivity and Rn capacity values were of the same magnitude, validating the reasonable choice of the force field parameters used in this study.

In order to establish a correlation between the structural features of COFs and their performance in capturing Rn, we conducted a detailed analysis of the spatial distributions of Rn in the top five candidates. This allowed us to identify the most preferential binding sites for Rn. Interestingly, our findings, as illustrated in Figure 2a–e, revealed a distinctive and fundamental structural feature shared by the dominant Rn capturing sites in different COF candidates. Specifically, these sites were constructed by well-defined cavities that were formed by the protruding groups of two opposite linkers of the COFs. Morphologically, these cavities had an approximately spherical inner space with a diameter of ~5.58 Å to 6.21 Å and a volume of ~106.76 Å^3^ to 123.56 Å^3^, demonstrating an excellent stereo shape matching the spherically shaped Rn (4.17 Å). This observation is consistent with the principle that cavities with geometries matching those of Rn are more thermodynamically favorable for accommodating Rn atoms, due to the relatively strong vdW dispersion interactions. A similar phenomenon was also observed in the case of ZIF-7, which has a Rn-matching spherical cage with a diameter of ~4.38 Å and displays remarkable Rn selectivity and uptake capacity.

### 3.2. The Influence of Humidity

As previously discussed, moisture can significantly impact the Rn capture performance of 2D COFs (as illustrated in Figure 1a,b). To gain further insights into the effect of RH on radon selectivity and adsorption capacity, a comprehensive investigation was carried out, as presented in Figure 3. The molar ratio of Rn/N_2_/O_2_ was maintained at 0.001/0.776/0.209, while the RH values were varied from 0 to 1.0.

Overall, the Rn selectivity and adsorption capacity of all five COF candidates exhibited a decreasing trend with increasing RH. As shown in Figure 3a, at lower RH levels (i.e., <0.3), the Rn selectivity of all five candidates demonstrated a rapid decline, whereas the rate of decrease slowed down considerably when RH exceeded 0.3. Similarly, the changes in Rn capacity also followed a similar pattern (Figure 3b). Nevertheless, it is worth noting that the extent of influence exhibited individual variations. For instance, linker105_C_linker13_C_kgm, which was ranked first at RH = 0.0, experienced the most significant impact on both Rn selectivity and Rn capacity (as shown in Figure 3). Conversely, linker101_C_linker11_C_kgm, which was ranked last at RH = 0.0, outperformed the other four candidates in terms of Rn selectivity when RH > 0.3 (Figure 3a). Additionally, its Rn capacity gradually surpassed the other three candidates as RH increased. At the highest RH of 1.0, its performance eventually reached a comparable level with that of linker105_C_linker13_C_kgm (Figure 3b). Overall, as RH increased beyond 0.3, all five candidates demonstrated a similar *APS_Rn_* performance.

To elucidate the molecular mechanism underlying the impact of RH on the Rn capturing performance of COFs, we investigated the Rn uptake behavior of two COF candidates, linker105_C_linker13_C_kgm and linker101_C_linker11_C_kgm, under different humidity conditions (RH = 0.0, 0.5, and 1.0) (Figure 4). These candidates were chosen because they exhibited distinct responses to changes in RH, with a much stronger influence observed on linker105_C_linker13_C_kgm than linker101_C_linker11_C_kgm.

At RH = 0.0, both candidates exhibited two major Rn binding sites (defined as site i and site ii, Figure 4a). With increasing RH, a significant decrease in Rn trapping events was observed in binding site i of linker105_C_linker13_C_kgm, accompanied by a remarkable increase in water trapping events (Figure 4a). In contrast, the changes were less significant in linker101_C_linker11_C_kgm (Figure 4c). At RH = 1.0, the Rn trapping events largely disappeared in linker105_C_linker13_C_kgm (Figure 4a, top panel), while mostly remaining in linker101_C_linker11_C_kgm (Figure 4c, top panel). Meanwhile, the water trapping events extensively appeared in linker105_C_linker13_C_kgm (Figure 4a, bottom panel), while they were less frequent in linker101_C_linker11_C_kgm (Figure 4c, bottom panel). Interestingly, the changes in binding site ii were much milder than those in binding site i for both candidates (Figure 4a,c). These results suggest that the Rn and water trapping events exhibit a reverse correlation in the same binding site, and that the extent to which RH affects the Rn trapping performance of the two candidates depends mainly on binding site i.

The distribution of adsorption energy of water/Rn in binding site i/site ii (Figure 4b,d) provides insight into these phenomena. In binding site i of linker105_C_linker13_C_kgm (Figure 4b left), the distribution of water adsorption energy clearly showed a leftward shift (toward the low-energy region) in contrast to that of Rn, indicating that replacing Rn with water could be thermodynamically favorable. In contrast, binding site ii (Figure 4b right) exhibited a reverse trend, indicating that trapping Rn is more thermodynamically favorable than trapping water. Thus, RH had a more dramatical influence on the trapping event of binding site i than it did on that of site ii. In binding site i of linker101_C_linker11_C_kgm (Figure 4c left), the situation was significantly different: the distribution of water adsorption energy exhibited a considerable overlap with that of Rn, and even showed a slightly rightward shift (toward the high energy region), indicating that replacing Rn with water may not always be thermodynamically favorable. In binding site ii (Figure 4c right), the Rn adsorption energy distribution exhibited a leftward shift in contrast to that of water, suggesting that Rn trapping is more thermodynamically favorable. Therefore, the main reason why RH has a much stronger influence on linker105_C_linker13_C_kgm (especially the binding site i) than on linker101_C_linker11_C_kgm is likely due to the leftward shift in the water adsorption energy distribution that contrasts that of Rn (Figure 3 and Figure 4).

Moreover, we conducted an analysis of the contributions of vdW and electrostatic interactions to the overall adsorption energy of adsorbate molecules, namely Rn and water, being trapped in various binding sites. As Rn is an inert gas, it is represented by a LJ particle, and therefore, the electrostatic interactions have no impact on Rn adsorption in all cases. The results presented in Table 2 show that, in all binding sites, the contribution of the vdW part to the total adsorption energy of water is lower than that of Rn. This is not surprising, given that the LJ potential well depth of Rn is lower than that of the oxygen atom in water.

Notably, in binding site i of linker105_C_linker13_C_kgm, a water molecule, on average, can acquire 3.41 kcal/mol energy from electrostatic interactions, which is sufficient to offset its lower energy gain from vdW interactions compared to the acquisition by Rn (water vs. Rn: 9.81 kcal/mol vs. 10.73 kcal/mol) (Table 2). Remarkably, this resulted in a substantial leftward shift in the adsorption energy distribution of water, relative to that of Rn (Figure 4b left). However, in the other three types of binding sites (including the binding site ii in linker105_C_linker13_C_kgm and binding sites i and ii in linker101_C_linker11_C_kgm), the energy gains from electrostatic interactions for a water molecule were insufficient to compensate for its lower energy gain from vdW interactions compared to those for Rn (Table 2), causing the adsorption energy distribution of water to shift to the right relative to that of Rn (Figure 4b right and Figure 4d). Consequently, the electrostatic interactions play a critical role in determining the overall adsorption energy distribution of water, which may ultimately govern the influence of RH on the Rn-selective capturing behavior.

The structural features of binding site i in linker105_C_linker13_C_kgm enable it to provide greater energy gains from electrostatic interactions with water. As depicted in Figure 5a, six protruding hydroxyl groups collectively point towards the interior space of the binding cavity, resulting in a higher polarization of electrostatic potential (ESP) surfaces of the cavity. This polarization creates a more favorable environment for electrostatic interactions with water. However, the spatial distributions of polar groups in the other three binding sites are less conducive to electrostatic interactions. For example, in binding site ii of linker105_C_linker13_C_kgm, hydroxyl groups point outward from the cavity, and in binding site i and site ii of linker101_C_linker11_C_kgm, amino groups are arranged in a planar-like manner. These distributions cannot induce significant changes in the polarization of the ESP surfaces of the cavities, resulting in less favorable electrostatic interactions with water (Figure 5a,b).

### 3.3. The Influence of the Rn Molar Fraction

Figure 6a,b illustrates the variations in Rn selectivity and uptake (adsorption) as a function of Rn molar fraction. In practice, the extremely low concentration (~10^−11^ ppb) is often beyond the numerical limit of most GCMC software (single-precision floating-point format). Hence, a relatively high gradient of molar concentration was employed to investigate the impact of gas concentration variations on nanoporous material performance [13,14]. Therefore, GCMC simulations were conducted under five different molar fractions, *X_Rn_* (*X_Rn_* = 0.00001, 0.00005, 0.0001, 0.0005, and 0.001), where moisture represents 0.5 (*X_water_* = 0.014) at ambient temperature, and the molar ratio of $X_{N_{2}}/X_{O_{2}}=0.776/0.209$.

As shown in Figure 6a, the Rn selectivity of the five COFs exhibited some fluctuations, but linker101_C_linker11_C_kgm displayed the highest Rn selectivity across all tested Rn molar fractions. However, the Rn uptake capacity of the five COFs exhibited a positive correlation with the Rn molar fraction. Specifically, when *X_Rn_* was below 0.0005, the Rn uptake capacity of linker105_C_linker13_C_kgm, linker107_C_linker13_C_kgm, and linker99_C_linker13_C_kgm was almost at the same level, while linker101_C_linker11_C_kgm and linker99_C_linker11_C_kgm had a comparable Rn capacity, albeit slightly lower than the former three candidates. At the highest Rn molar fraction (*X_Rn_* = 0.001), the differences in the performances of the COFs became apparent, with the Rn uptake capacity order being linker105_C_linker13_C_kgm > linker107_C_linker13_C_kgm > linker99_C_linker13_C_kgm = linker99_C_linker11_C_kgm > linker101_C_linker11_C_kgm.

### 3.4. The Influence of Pressure

We also evaluated the Rn-capture isothermal adsorption curves of screened five COFs under RH = 0.5 at room temperature. Figure 7 depicts the results of these tests, which were conducted at *X_Rn_* = 0.001, with pressures ranging from 5 to 140 kPa. As shown in Figure 7a, Rn selectivity varied with increasing pressure, with linker101_C_linker11_C_kgm exhibiting the most significant performance. At the highest pressure (Figure 7b), the performance order was linker105_C_linker13_C_kgm > linker107_C_linker13_C_kgm = linker109_C_linker13_C_kgm > linker99_C_linker11_C_kgm > linker101_C_linker11_C_kgm. These results demonstrate the potential of screened COFs as effective Rn capture agents in various environmental settings.

## 4. Conclusions

In this study, we investigated the potential of 8641 two-dimensional covalent organic frameworks (COFs) to capture radon (Rn) under humid air conditions. To achieve this, high-throughput GCMC simulations were used to screen the COFs based on their Rn capture capability from a quaternary mixed gas with a molar ratio of Rn/N_2_/O_2_/H_2_O of 0.001/0.776/0.209/0.014 at ambient temperature and pressure. After thorough evaluation, we identified five COFs, namely linker105_C_linker13_C_kgm, linker101_C_linker11_C_kgm, linker107_C_linker13_C_kgm, linker99_C_linker11_C_kgm, and linker99_C_linker13_C_kgm, as the top candidates for Rn capture. The analysis of structure–performance relationships suggested that the well-defined cavities in COFs, having a diameter that matches that of Rn, represent a very crucial prerequisite for competitive Rn capturing sites because the excellent steric match between the cavity and Rn could maximize their vdW dispersion interactions for the adsorption energy.

We further evaluated the Rn capture performance of the selected COFs under different humidity conditions (RH = 0.0, 0.5, and 1.0). The results showed that both the Rn selectivity and adsorption capacity of all five COFs decreased with increasing RH, with a more pronounced effect observed at lower humidity levels (RH < 0.3). Interestingly, the impact of humidity on the performance of COFs exhibited individual differences. For instance, RH had a greater influence on the performance of linker105_C_linker13_C_kgm than it did on the performance of linker101_C_linker11_C_kgm. The underlying molecular mechanism is attributed to the architecture of polar groups in the binding sites. For instance, in binding site i of linker101_C_linker11_C_kgm, there are six protruding hydroxyl groups that collectively point to the interior space of the cavity, inducing a notable-polarization ESP surface that offers more favorable electrostatic interactions for water. However, in the other three binding sites, the spatial distributions of the polar groups point outward from the cavity (e.g., the hydroxyl groups in binding site ii of linker105_C_linker13_C_kgm) or are arranged in a planar-like manner (e.g., the amino groups in binding site i and site ii of linker101_C_linker11_C_kgm), which cannot induce the remarkable-polarization ESP surfaces of the cavities and are hence less favorable for electrostatic interactions with water. This is the main molecular mechanism by which RH impacts the Rn capturing performance of COFs. Lastly, we evaluated the influence of Rn molar fractions and pressures on the performance of the COFs. It should be noted that, under equal conditions (at RH = 0.5, 1 bar, 298 K, and *X_Rn_* = 0.001), the five screened COFs exhibited slightly lower Rn capacity (ranging from ~0.73 to ~0.96 mol/kg) but much better Rn selectivity (ranging from ~244.1 to ~392.5) compared to ZIF7-Im-1 (capacity: ~1.04 mol/kg; selectivity: ~64.1). To the best of our knowledge, experimentally, ZIF7-Im-1 may remain the record holder for both the highest Rn uptake capacity and selectivity so far [14]. Given their relatively stable Rn capturing performance in moist air, these COFs could potentially serve as promising candidates for Rn capture, which deserves further experimental verification.

## Figures and Tables

**Figure 1 nanomaterials-13-01532-f001:**
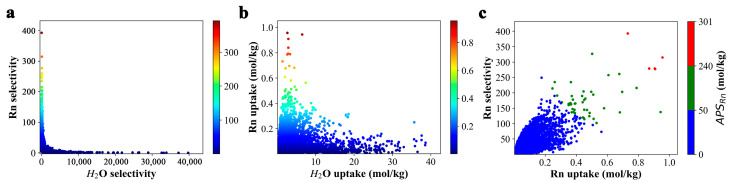
High-throughput computational screening of Rn-capturing 2D COFs. Scatter plots of (**a**) Rn selectivity vs. water selectivity, (**b**) Rn uptake capacity vs. water uptake capacity, and (**c**) APS values of COFs in moist air (comprising a mixed gas of Rn/N_2_/O_2_/H_2_O, with a molar ratio of 0.001/0.776/0.209/0.014) at ambient temperature (298 K) and 1 bar; RH = 0.5.

**Figure 2 nanomaterials-13-01532-f002:**
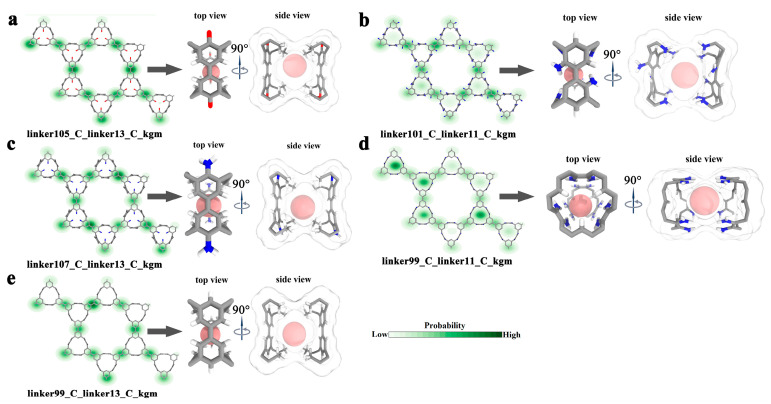
Rn capturing sites of top 5 COFs. Radon adsorption density and the most preferential Rn capturing site(s) of (**a**) linker105_C_linker13_C_kgm, (**b**) linker101_C_linker11_C_kgm, (**c**) linker99_C_linker13_C_kgm, (**d**) linker107_C_linker17_C_kgm, and (**e**) linker99_C_linker11_C_kgm. Each atom is in a different color: carbon is in silver, hydrogen is in white, oxygen is in red, nitrogen is in blue, and Rn is in pink.

**Figure 3 nanomaterials-13-01532-f003:**
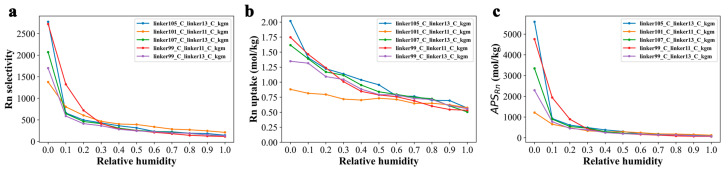
Rn capturing performance of top 5 COFs under different RH conditions. (**a**) Selectivity, (**b**) adsorption capacity, and (**c**) *APS_Rn_* of top 5 2D COFs for radon capture at a X_Rn_ of 0.001, temperature of 298 K, pressure of 1 bar, and RH between 0 and 1.0.

**Figure 4 nanomaterials-13-01532-f004:**
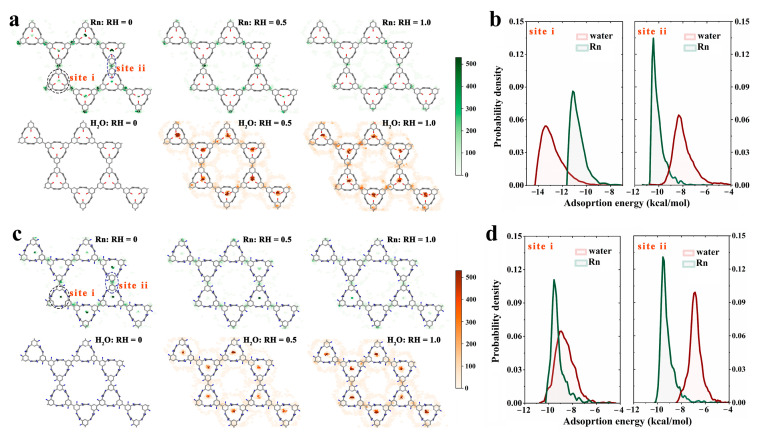
The frequency of Rn (top panel) and water (bottom panel) capture events in binding sites i and ii under RH of 0, 0.5, and 1.0, respectively, as well as the adsorption energy distributions for Rn (green) and water (deep red) trapped in binding site i (left) and site ii (right). (**a**,**b**) denote the linker105_C_linker13_C_kgm case, while (**c**,**d**) denote the linker101_C_linker11_C_kgm case. The green/red color bar indicates the number of Rn/water capture events.

**Figure 5 nanomaterials-13-01532-f005:**
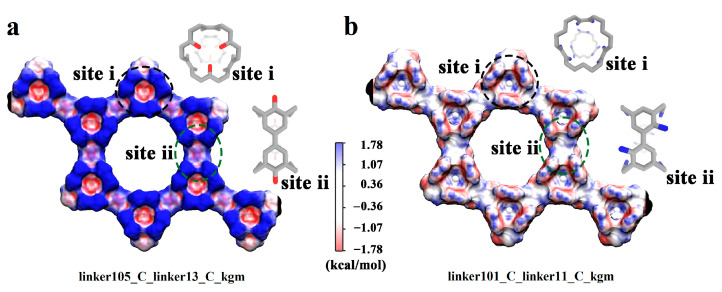
The surface electrostatic potential of binding site i and site ii in (**a**) linker105_C_linker13_C_kgm and (**b**) linker101_C_linker11_C_kgm, respectively.

**Figure 6 nanomaterials-13-01532-f006:**
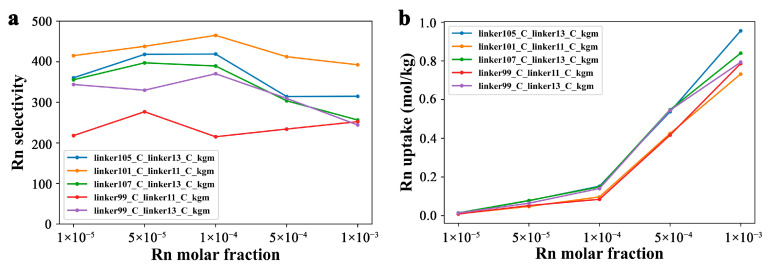
Rn capturing performance of top 5 COFs under different Rn molar factions. (**a**) Rn selectivity and (**b**) adsorption capacity of top 5 COFs under different Rn concentrations (*X_Rn_* = 0.00001, 0.00005, 0.0001, 0.0005, and 0.001) at the pressure of 1 bar, ambient temperature of 298 K, and molar ratio of $X_{N_{2}}/X_{O_{2}}=0.776/0.209$.

**Figure 7 nanomaterials-13-01532-f007:**
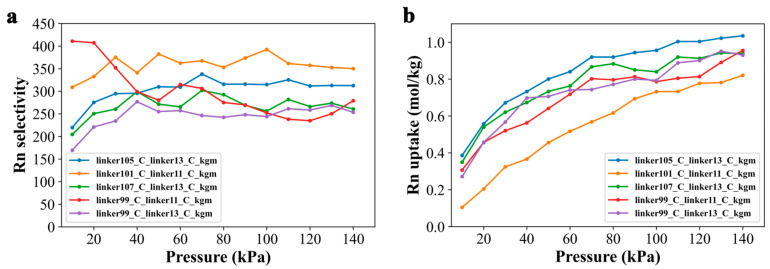
Rn capturing performance of top 5 COFs under different pressures. (**a**) Selectivity and (**b**) adsorption capacity of 5 COFs for radon at the *X_Rn_* of 0.001 and 298 K, with the mixed gas of Rn/N_2_/O_2_/H_2_O at a molar ratio of 0.001/0.776/0.209/0.014.

**Table 1 nanomaterials-13-01532-t001:** The Lennard-Jones parameters and atomic charges for adsorbates (including Rn, N_2_, O_2_ and H_2_O) in the GCMC simulations ^a^.

Molecules	Geometry	Atoms	*σ* (Å)	*ε/k_b_* (K)	q (e^−^)
Rn			4.53	272.243	0.000
N_2_	*d*_NN_ =1.1000 Å	N-N_2_	3.31	36.000	−0.482
		COM-N_2_	0.00	0.000	0.960
O_2_	*d*_OO_ =1.2100 Å	O-O_2_	3.05	54.400	−0.112
		COM-O_2_	0.00	0.000	0.224
H_2_O (TIP3P)	*d*_OH_ = 0.9572 Å ∠HOH = 104.52°	H-H_2_O	0.00	0.000	0.417
		O-H_2_O	3.15	76.540	−0.834

^a^ d_NN_, d_OO_, and d_OH_ represent the interatomic distances of the N-N, O-O and O-H bonds in nitrogen, oxygen, and water, respectively. ∠HOH denotes the angle formed by H-O-H in water. COM indicates the center of mass of the adsorbate molecule.

**Table 2 nanomaterials-13-01532-t002:** The average adsorption energy (kcal/mol) of an adsorbate (Rn/water) molecule being captured in binding site i and site ii in linker105_C_linker13_C_kgm and linker101_C_linker11_C_kgm, which is denoted as ΔG_total_
*^a^*.

COF	Sites	Adsorbates	ΔG_elec_	ΔG_vdW_	ΔG_total_
linker105_C_linker13_C_kgm	site i	Rn	N/A	−10.73	−10.73
		water	−3.41	−9.81	−13.22
	site ii	Rn	N/A	−10.05	−10.05
		water	−1.13	−7.12	−8.25
linker101_C_linker11_C_kgm	site i	Rn	N/A	−9.25	−9.25
		water	−0.51	−8.41	−8.92
	site ii	Rn	N/A	−9.08	−9.08
		water	−0.41	−6.56	−6.97

*^a^* ΔG_elec_ and ΔG_vdW_ represent the contribution of the electrostatic and vdW interactions to the total adsorption energy (ΔG_total_), respectively.

## Data Availability

The data presented in this study are available on request from the corresponding author.

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
