# Peer review of "High-Throughput Computational Screening of Two-Dimensional Covalent Organic Frameworks (2D COFs) for Capturing Radon in Moist Air"

_nanomaterials, 2023, doi:10.3390/nano13091532_

Round 1
Reviewer 1 Report (Previous Reviewer 3)
-
Author Response
We sincerely appreciate the referee’s expertise and insightful comments, which are very helpful for revising and improving our manuscript.
Reviewer 2 Report (Previous Reviewer 2)
Comments from Reviewer
Title: High-throughput Computational Screening of Two-Dimensional Covalent Organic Frameworks (2D-COFs) for Capturing Radon in Moist Air
The current form's presentation of methods and scientific results is satisfactory for publication in the Nanomaterials journal. Congratulations on a great job. The author has made a substantial improvement in this article. Overall the manuscript improved. The minor and significant drawbacks to be addressed can be specified as follows:
1. Line 377. The influence of Pressure ---> The influence of pressure.
2. There is no comparison of the abilities of the tested materials with other "real" materials.
Sincerely,
The reviewer.
The current form's presentation of methods and scientific results is satisfactory for publication in the Nanomaterials journal. Congratulations on a great job. The author has made a substantial improvement in this article. Overall the manuscript improved.
Author Response
The minor and significant drawbacks to be addressed can be specified as follows:
- Line 377. The influence of Pressure ---> The influence of pressure.
Author reply: Fixed. (in blue)
- There is no comparison of the abilities of the tested materials with other "real" materials.
Author reply: Thank you for the excellent suggestion. We have included more comparisons of the Rn capture performance of the screened top 5 2D COF candidates with ZIF7-Im-1. To the best of our knowledge, experimentally, ZIF7-Im-1 may still hold the record for both the highest Rn uptake capacity and selectivity so far. Under equal conditions (at RH = 0.5, 1 bar, 298 K, and XRn = 0.001), the screened five COFs exhibited slightly lower Rn capacity (ranging from ~0.73 to ~0.96 mol/kg) but much better Rn selectivity (ranging from ~244.1 to ~392.5) compared to ZIF7-Im-1 (capacity: ~1.04 mol/kg; selectivity: ~64.1). (in blue)

Reviewer 3 Report (Previous Reviewer 1)
Accept
Author Response
We sincerely appreciate the referee’s expertise and insightful comments, which are very helpful for revising and improving our manuscript.
This manuscript is a resubmission of an earlier submission. The following is a list of the peer review reports and author responses from that submission.
Round 1
Reviewer 1 Report
Detailed comments:
1. The English of the text should be checked
2. The novelty of the manuscript is complete missing
3. For all parameters used in the equations indicate what represent
4. For all abbreviations or notation indicate the complete name
5. For unit of measure used the S.I., please check in all manuscript.
6. For all operational conditions must be indicated, amount, concentration, time, speed rotation
7. For all reagents or chemicals used must be indicated manufacturer, purity, concentration, amount
8. All equipment and tools used in this study should be described in detail or further information should be provided (manufacturer, type, operational conditions, etc.)
9. More Conclusions with the best obtained values
10. Other more potential application of the materials used must be indicated in the Conclusion part
Reviewer 2 Report
Comments from Reviewer
Title: High-throughput Computational Screening of Two-Dimensional Covalent Organic Frameworks (2D-COFs) for Capturing Radon in Moist Air
The current form's presentation of methods and scientific results is unsatisfactory for publication in the Nanomaterials journal. The minor and significant drawbacks to be addressed can be specified as follows:
1. Abstract. I suggest writing an abstract with descriptive and concise information about this research. A standard definition is: “An abstract is a concise summary of an experiment or research project. It should be brief -- typically under 200 words. The purpose of the Abstract is to summarize the research paper by stating the purpose of the research, the experimental method, the findings, and the conclusions.” In my opinion, the Abstract is too extended.
2. Keywords. Do not repeat the words in the title to the “keywords”.
3. Line 70. Smith, R. M. et al. ---> Smith et al. See also “Berend S. et. al.”. Others?
4. Lines 84 and 85. Why this composition? Please, explain this choice.
5. Fig. 1 and Fig. S1. Number of COFs???
6. Fig. 2 and Tab. S1. (c) linker16? (d) linker89? (e) linker81?
7. Line 226. The influence of Humidity ---> The influence of humidity
8. References. Literature should also be standardized: the size of letters in the titles of journals, initials of names, the size of letters in the titles of articles. See, for example: (i) [1] “-+”? (ii) [4] doi:Doi??? (iii) [7] Pages? (iv) [24] “a Full” small letter “a” ? (v) [25] acs ---> ACS
9. There is no comparison of the abilities of the tested materials with other materials.
Sincerely,
The reviewer.
Reviewer 3 Report
The idea for a study seems quite interesting. However, some issues should be adressed prior to publication.
1) The authors should clearly define the objectives of the study and indicate the elements of scientific novelty.
2) Why did the authors limit the research to relative humidity not exceeding 80% (lines 86-87)?
3) The authors refer to a work with interaction parameters published more than 30 years ago. Much progress has been made in modelling since then. It also applies to models for radon. From the point of view of the subject, it is important to use a force field for radon that is as realistic as possible.
See for example
http://dx.doi.org/10.1021/acs.jced.5b01002
The authors should check at least for selected systems to what extent the interaction parameters for Rn affect the obtained results!
4) Which water molecule model (SPC, TIP3, TIP4 etc.) was used? Were N2 and O2 molecules modelled as single LJ centres or multi-centre molecules?
5) The mole fractions of Rn assumed by the authors seem very high. Are there reports in the literature indicating that radon in the air in buildings can be present in such a high concentration?
6) In my opinion, the authors focused too much on selectivity. From a practical point of view, the amount of captured radon is crucial, regardless of whether and in what amount other air components are adsorbed.
7) Fig. 1 and its discussion – The authors also should show Rn selectivity and uptake versus water adsorption amount. This should enrich the discussion and indicate potential additional regularities. It seems to me very likely that the observed low selectivities of Rn can be related to the volume filling of the pores by H2O molecules. However, it is interesting how this affects the absolute amount of adsorbed Rn.
8) Section 4 – The authors should also show the H2O adsorption amount. I suspect that the volume filling of pores by water molecules does not occur for selected systems under the studied conditions. The authors should also show and discuss the influence of humidity for at least one system in which volume pore filling occurs.
Minor technical comments:
9) Some names in the text are accompanied by initials – see lines 70 and 76 for example. However, the authors are not consistent and names without initials appear elsewhere. The latter way is usually used in scientific texts.
10) Line 83 ‘giant canonical Monte Carlo’ – perhaps ‘grand canonical Monte Carlo’?
11) What does 'cycle' mean (lines 91 and 92)?
12) Eq. (1) - The authors do not use the selectivity coefficient defined in this way (dedicated to binary mixtures). They should provide a definition of the parameter actually determined for the studied multi-component mixtures.
13) Fig. 1 – The authors also visualised the quantity called ‘number of COFs’. Its meaning is unclear!
14) Fig. 2 – The colour scale probably represents the density of Rn, but the authors did not specify units of this quantity.
15) Line 188 – Fig. 3 does not present selectivity.
14) Lines 202, 208, 224 and 251 – ‘298 K’ instead of ‘298 k’!
16) Line 208 – ‘T =’ instead of ‘t =’!
17) Captions of Figs 3 and 4 – The authors should provide all parameters determining the composition of the gas phase.
18) Refs [1] and [6] – the lack of correct page range.
19) Refs [17] and [18] – the lack of page range.